# Fabrication of PCDTBT Conductive Network via Phase Separation

**DOI:** 10.3390/ma14175071

**Published:** 2021-09-04

**Authors:** Jianwei Xu, Zhiming Liu, Lei Jing, Jingbo Chen

**Affiliations:** School of Materials Science & Engineering, Zhengzhou University, Zhengzhou 450002, China; LZM18438612316@163.com (Z.L.); aswh26357856@163.com (L.J.)

**Keywords:** PCDTBT, PEG, polymer blends, phase separation, conductive network

## Abstract

Poly[N-9′-hepta-decanyl-2,7-carbazole-alt-5-5-(4′,7′-di-2-thienyl-2′,1′,3′-benzothiadiazole)] (PCDTBT) is a stable semiconducting polymer with high rigidity in its molecular chains, which makes it difficult to organize into an ordered structure and affects the device performance. Here, a PCDTBT network consisting of aggregates and nanofibers in thin films was fabricated through the phase separation of mixed PCDTBT and polyethylene glycol (PEG). Using atomic force microscopy (AFM), the effect of the blending conditions (weight ratio, solution concentration, and molecular weight) and processing conditions (substrate temperature and solvent) on the resulting phase-separated morphologies of the blend films after a selective washing procedure was studied. It was found that the phase-separated structure’s transition from an island to a continuous structure occurred when the weight ratio of PCDTBT/PEG changed from 2:8 to 7:3. Increasing the solution concentration from 0.1 to 3.0 wt% led to an increase in both the height of the PCDTBT aggregate and the width of the nanofiber. When the molecular weight of the PEG was increased, the film exhibited a larger PCDTBT aggregate size. Meanwhile, denser nanofibers were found in films prepared using PCDTBT with higher molecular weight. Furthermore, the electrical characteristics of the PCDTBT network were measured using conductive AFM. Our findings suggest that phase separation plays an important role in improving the molecular chain diffusion rate and fabricating the PCDTBT network.

## 1. Introduction

Conjugated polymers have been extensively studied, as they show great potential in achieving large-area, lightweight, flexible electronics, such as organic solar cells (OSCs) and organic field-effect transistors (OFETs) [1,2,3,4,5]. Poly[N-9″-hepta-decanyl-2,7-carbazole-alt-5-5-(4′,7′-di-2-thienyl-2′,1′,3′-benzothiadiazole)] (PCDTBT), one of the most prominent conjugated polymers, has attracted a great deal of attention due to its thermal stability, low optical band-gap, and ambipolar property [6,7,8,9,10]. However, compared with other conjugated polymers, such as poly(3-hexylthiophene) (P3HT), the high rigidity of PCDTBT molecular chains makes it more difficult to organize into an ordered structure [11,12].

It is well-known that the performance of electronic devices was determined by the morphology and structure of conjugated polymers [13,14,15,16]. As the charge carrier transfers faster along conjugated back-bones and π-stacking direction, crystals exhibit better charge transport properties than the amorphous zone [17,18,19]. Moreover, the orientation of micro- and nano-scale ordered structures also plays an important role in charge transport [20,21,22]. Up to now, many approaches have been carried out to control the ordered structures of conjugated polymers as well as their properties, including crystallization from solution, annealing, phase separation, etc. [23,24,25,26]. When two immiscible polymers are mixed, the most frequent result is a system that exhibits almost total phase separation, which significantly influences their final morphology [27,28,29,30,31,32]. This can be explained in terms of the reduced combinatorial entropy that results from mixing two types of polymer chains. Two mechanisms, the nucleation-growth mechanism and the spinodal decomposition mechanism, have been developed to explain the corresponding phase separation behavior in the thermodynamically metastable and unstable regions of the phase diagram [33,34]. Han [35], Liu [36], and Ginger [37] reported the influence of phase separation on P3HT crystallization. However, phase-separated morphology and structure are sensitive to many parameters, such as the polymer molecular weight, the blend weight ratio, the solution concentration, etc. [38,39,40,41]. High-performance, semiconducting architectures can be obtained through blending a conjugated polymer with another semi-crystalline polymer, based on crystallization-driven phase separation [42].

Conjugated polymers’ films or devices were usually fabricated using the spin-coating technique, where a mixed solution of the polymer and a volatile solvent is spun at high speeds on a solid substrate. In a fast-drying process accompanied by phase separation, polymer chains are transformed into ordered structures under conditions far away from thermodynamic equilibrium, which is similar to the process that occurs during the spin-coating of mixed polymer solutions. It is unclear how polymer chains are transported to the growing aggregates over long distances of a few micrometers in the extremely short time available (about a few seconds) in such evaporating, thin films. To explore how phase separation and crystallization affect the formation of the ordered structures of conjugated polymers, this work blended PCDTBT with a well-studied model polymer, PEG [43,44,45] (PEG is a kind of flexible polymer with a relatively strong crystallization ability and it can be easily removed by solvents), with chlorobenzene as a common solvent. The conductivity of the phase-separated morphologies of PCDTBT was investigated using conductive atomic force microscopy (CAFM).

## 2. Experimental Section

The chemical structures of PCDTBT and PEG are shown in Appendix A. Two PCDTBT samples (*M*_n_ = 16.2 and 30.0 kg/mol) were purchased from Ossila and 1-material Inc., respectively. Five PEG materials (*M*_n_ = 1, 4.6, 20, 35, and 100 kg/mol) were obtained from Aldrich Chemical Co. The chlorobenzene and chloroform were analytical grade, and they were used without any further purification.

PCDTBT and PEG solutions with different concentrations (0.1–3.0 wt%) were prepared by dissolving each polymer in chlorobenzene; the solutions were then annealed at 70 °C for 6 h and 1 h, respectively [46,47]. A series of blend solutions were produced by mixing PCDTBT and PEG solutions at weight ratios of 2:8, 4:6, 5:5, 6:4, 7:3, and 8:2. To prepare PCDTBT/PEG blend films, mixed solutions were spin-coated onto silicon wafers (P100 type) using a KW-4A spin-coater. The spin speed and time were 3000 rpm and 60 s, respectively. Silicon substrates were treated by UV ozone for 1 h before being placed on the spin-coater, heated by a homemade heater, and kept at the spin-coating temperature (*T*_sp_) for 5 min before coating. All experiments were performed under a nitrogen environment.

The surface morphologies of the resulting blend films were collected using Bruker Dimension Icon atomic force microscopy (AFM) in ScanAsyst mode under nitrogen, and an Olympus BX-51 optical microscope equipped with a Linkam THMS 600 hot stage. To facilitate a clear mapping of the surface topography and electrical properties, PCDTBT/PEG blend films were washed by dipping the samples into hot acetone (e.g., at 50 °C for 5 s) to selectively remove the PEG regions. The conductivity of the PCDTBT structures was collected in the PeakForce Tunneling AFM mode (PF-TUNA).

## 3. Results and Discussion

In Figure 1a, a topographical image of a 0.5 wt% PCDTBT (*M*_n_ = 16.2 kg/mol)/PEG (*M*_n_ = 35 kg/mol) (1:1) blend film exhibits a morphology of islands and peninsulas. In order to investigate the structures of the PCDTBT, a washing experiment was carried out by using acetone (a good solvent only for PEG) to selectively remove the PEG phase (see Figure 1d). After washing, a network consisting of islands connected by nanofibers was obtained [35,48]. Height distributions for the cross-sections—indicated by the white dashed lines in Figure 1a,d—are shown in Figure 1b,e, respectively. It is worth emphasizing that the PCDTBT aggregate structures (both islands and nanofibers) can be revealed through a selective washing procedure, as shown in the height distributions. Furthermore, one can observe nanofibers with dimensions of about 20 nm in width and 5 nm in height. As can be seen from the histogram of the height distribution shown in Figure 1c,f, the mean height difference between the PCDTBT aggregate structures and the PCDTBT nanofibers is about 25 nm. This value is larger than the lamellae thickness of PEG (~10 nm), indicating that PCDTBT molecules did not adsorb flatly on the substrate but rather were aggregated with each other.

To better understand the phase separation and the formation of the PCDTBT network, we first investigated the influence of the weight ratio [39,49,50]. Figure 2a–f shows the AFM height images of blend films with different weight ratios (PCDTBT/PEG = 2/8, 4/6, 5/5, 6/4, 7/3, and 8/2). The size of each AFM image is 3 × 3 μm^2^ (Except for Figure 2d, where the size is 35 × 35 μm^2^). Phase separation can be observed in Figure 2a–e. With the increase in the PCDTBT component, a phase-separated structure transition from a sea-island to a continuous structure occurs, which is indicative of a change from a nucleation-growth mechanism to a spinodal decomposition mechanism in the blend system [34]. Further, in the case of 8/2 (see Figure 2f), a crystal appears on the surface, which might be explained by the fact that excessive PCDTBT molecules tend to form a film and PEG crystallizes on the surface of the PCDTBT matrix [35].

AFM height images of the films after the selective removal of the PEG using acetone are displayed in Figure 2a1–e1. After removing the PEG phase from the blends, a network consisting of PCDTBT aggregates and nanofibers is revealed, while no fibers (especially nanofibers) are observed in Figure 2f1. In addition, the pure PCDTBT thin film shows a similar morphology to that of Figure 2f1 (see Appendix A), indicating that no PCDTBT nanofibers exist in the initial solution. According to these results, considering that PEG is a kind of crystalline polymer with relatively higher flexibility than PCDTBT [6,23,43], it is suggested that the PCDTBT network is formed during the phase separation, which is driven by the incompatibility of PEG/PCDTBT and the crystallization of PEG.

Figure 2g–k shows the AFM height images of the washed films, which are prepared by blend films of different concentrations (*c* = 2.0–0.1 wt%, other concentrations can be seen in Appendix A). Significant differences in the morphology of PCDTBT aggregates and nanofibers were observed. As *c* decreased from 2.0 wt% to 0.1 wt%, both the height of the PCDTBT aggregates and the width of the nanofibers became smaller (see Figure 2m). It has been known that the timescale of phase separation increases in blend films that are spin-coated using more highly concentrated solution because of the higher viscosity of the solution [51]; this explains the increasing heights of the aggregates and widths of the nanofibers.

It is imperative to point out that the length of nanofibers decreased with decreasing *c*, probably due to the enhanced mobility of the PCDTBT molecules. Especially, for the lowest *c* (i.e., 0.1 wt%), only the PCDTBT nanofiber network rather than the PCDTBT aggregates were found in the washed film (Figure 2k). This morphological change may be caused by the absorption of a majority of the PEG molecules onto the hydrophilic oxidized silicon wafers, which may result in an adsorbed PEG monolayer rather than aggregates or crystals in PEG-rich domains [52]. The crystallization of PEG monolayers was predominately controlled by the diffusion of PCDTBT molecules leading to the shorter nanofibers [53,54,55].

Due to mobility and compatibility being sensitive to the molecular weights of polymers [34], PEG and PCDTBT with different molecular weights were blended here. Figure 3a–e shows the AFM height images of blend films, with weight ratios of 4:6 and the increasing molecular weight of PEG from 1 kg/mol to 100 kg/mol. A series of PCDTBT networks consisting of nanofibers and aggregates with different sizes and densities can be observed in the resulting films. The average height and density of the PCDTBT aggregates as a function of the PEG molecular weight were revealed (see Figure 3f). With the increasing molecular weight of the PEG, the incompatibility of the two blend components increases, causing that phase separation scale to increase and the PCDTBT aggregates to coarsen further to form a larger structure [34]. On the other hand, shorter PEG chains diffuse more rapidly, leading to PCDTBT aggregates becoming more dispersed [38].

Furthermore, the effect of the molecular weight of the PCDTBT on the phase-separated morphology was studied. Figure 3g,h show the AFM height images of the washed films with PCDTBT molecular weights of 16.2 kg/mol and 30.0 kg/mol (weight ratio = 6:4; the ratios of 4:6 are shown in Appendix A). The corresponding height distributions of the dashed lines in Figure 3g (red curve) and Figure 3h (green curve) are shown in Figure 3i. The molecular weight dependencies of the density and width of PCDTBT nanofibers are clearly resolved. The resulting porous patterns of lower *M*_n_ PCDTBT possessed a morphology with lower pore density and larger pore diameter. As low molecular weight PCDTBT expresses lower viscosity and higher diffusivity, the aggregates are more dispersed when induced by phase separation [56]. Meanwhile, the width of the nanofiber is larger in Figure 3g, which can be attributed to the decreased viscosity of the low molecular weight PCDTBT [38].

Temperature also plays an important role for the phase-separated structure, as it determines the thermodynamic state of the blend system. We thus explored the impact of the spin-coating temperature (*T*_sp_) on the morphology of films, which spin-coated from a mixed solution with a polymer concentration of *c* = 0.5 wt% and a mixing ratio of 6:4 (PCDTBT/PEG). When *T*_sp_ = 50 °C, the transition from a continuous to an island structure was observed. We quote the classical phase separation theory, considering that the continuous phase-separated morphology (Figure 4a) is driven by the spinodal decomposition mechanism [57]. With the increase in the temperature, the blend system is thrust into the metastable region, causing the phase-separated morphology of the island (Figure 4b) that is controlled by the nucleation-growth mechanism [57].

Since the time-span available for the phase-separation process is determined by the evaporation rate of the solvent [58], we chose chloroform—another good solvent for both polymers that has a faster volatilization rate than chlorobenzene—to prepare the blend solutions. When compared to the sample prepared using chlorobenzene as a solvent (Figure 4c), fewer nanofibers are observed in the sample obtained from the chloroform solution (Figure 4d). This is because rapid solvent evaporation occurs during the spin-coating stage, with less time available for phase separation, thus hindering the self-assembly of the PCDTBT molecules into nanofibers.

After the selective removal of PEG, the conductivity of the PCDTBT aggregates and nanofibers was investigated using the PeakForce TUNA mode AFM. As shown in Figure 4e,f, the morphology of the PCDTBT aggregates and nanofibers were all reproduced in the current map of the washed film taken at an applied voltage of 800 mV. That is, the conductivity of the PCDTBT aggregates and nanofibers was significantly higher than that of the substrates. Interestingly, the conductivity of the PCDTBT aggregates and nanofibers decays after heating to 160 °C for 30 min under a nitrogen atmosphere, due to the reduction of the ordered degree (see Figure 4h). As observed in the AFM phase images (see Figure 4i,j), the PCDTBT fiber-aggregate network showed the physical connection between aggregates and nanofibers. Therefore, the formation of interconnected pathways is desired to ensure highly efficient, long-range charge transport.

Based on all the information obtained from the above content, a schematic diagram of the PCDTBT network formation process is shown in Figure 4k. First, when dropping the blend solution onto a silicon substrate, PCDTBT and PEG components uniformly mix in droplets on the substrate. Then as the spin-coating method is manipulated to fabricate the thin film, the droplets are thrown off and the blends evenly distribute on the substrate. Meanwhile, with the evaporation of the solvent, phase separation occurs (driven by an interaction, such as the surface tension between two immiscible polymers) and subsequently nanofibers and aggregates are formed. Notice that when using deionized water as the solvent to remove the PEG phase of the blend film, the film can float on the water, indicating there is a PEG-rich layer between the substrate and the PCDTBT nanofibers. This can be explained by a better affinity between PEG and substrate.

## 4. Conclusions

In summary, a PCDTBT conductive network consisting of aggregates and nanofibers was successfully produced through phase separation of a PCDTBT/PEG blend. The network was revealed after the selective removal of the PEG regions by using acetone. The phase-separated morphologies of this blend system were influenced by the blending conditions (the weight ratio, solution concentration, and molecular weight) and processing conditions (the substrate temperature and solvent). A transition from a sea-island, phase-separated structure to a continuous structure was observed when the PCDTBT:PEG weight ratio was changed from 2:8 to 7:3 or when the substrate temperature decreased from 50 °C to 20 °C. Varying the solution concentration from 0.1 wt% to 3.0 wt% led to an increase in the film thickness and solution viscosity, causing an increase in both the PCDTBT aggregate height and the nanofiber width. Meanwhile, when the molecular weights of the PEG and PCDTBT were increased, both the diffusion rate of the molecular chains and the miscibility of the blends decreased, resulting in larger sizes of PCDTBT aggregates and denser nanofibers, respectively. However, no nanofibers were observed when using a solvent with a low boiling point (chloroform), as rapid solvent evaporation shortened the time available for nanofibers to be induced through phase separation. These results suggested that the PCDTBT network was induced during the stage of phase separation and was driven by the incompatibility of PEG/PCDTBT and the crystallization of the PEG. In addition, the conductivity of the PCDTBT structures obtained from the blend films was proved through CAFM, giving a bright prospect for its application in devices and also providing new insight in dealing with highly rigid conjugated polymers, such as PCDTBT.

## Figures and Tables

**Figure 1 materials-14-05071-f001:**
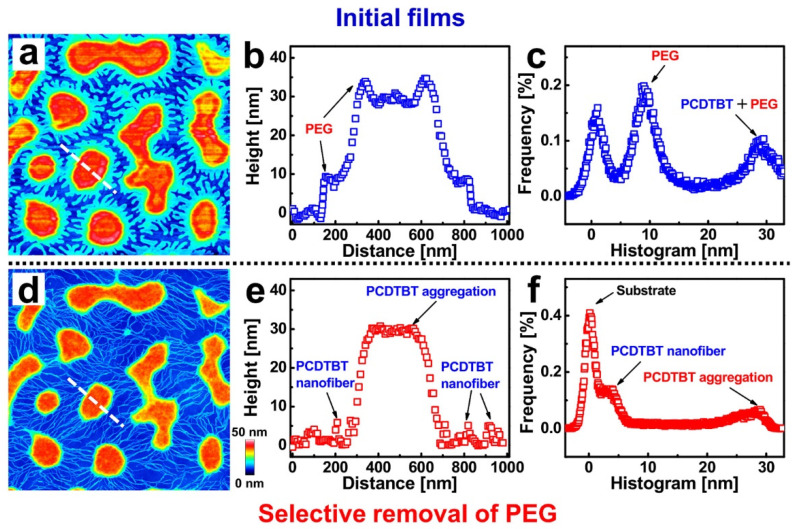
**Morphology of PCDTBT-PEG blend film.** (**a**,**b**) AFM topographical images of a typical PCDTBT/PEG (weight ratios of 5/5) blend film and the corresponding washed film after selective removal of PEG using acetone. (**b**,**e**) Height distributions along the white dashed lines in (**a**,**d**). (**c**,**f**) Height histograms obtained from the AFM images shown in (**a**,**d**). The size of each AFM image is 3 × 3 μm^2^.

**Figure 2 materials-14-05071-f002:**
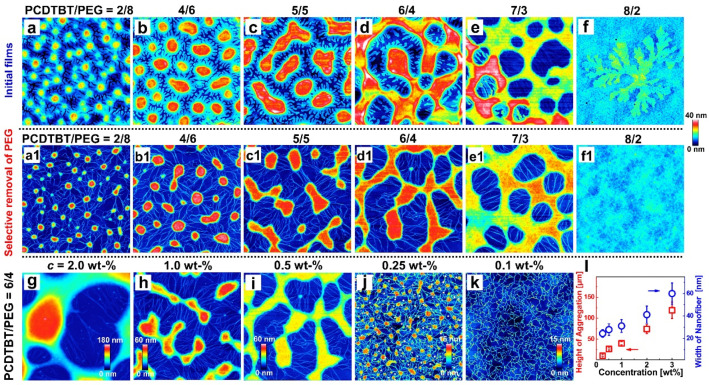
**Effects of weight ratio and concentration on the morphology of blend films.** AFM height images of (**a**–**f**) initial blend film and (**a1**–**f1**) washed films, each containing weight ratios of 2/8, 4/6, 5/5, 6/4, 7/3, and 8/2. AFM height images of washed films with a weight ratio of 6/4 in the presence of different concentrations (**g**) 2.0 wt%, (**h**) 1.0 wt%, (**i**) 0.5 wt%, (**j**) 0.25 wt%, and (**k**) 0.1 wt%. (**m**) The height of the aggregate structures (red) and the width of the nanofibers (blue) as a function of solution concentration.

**Figure 3 materials-14-05071-f003:**
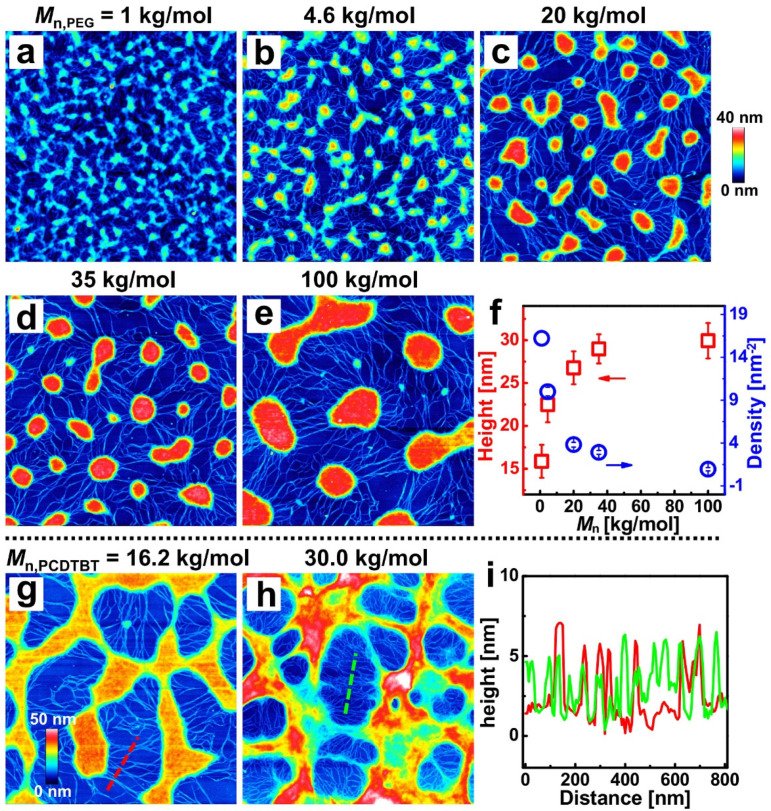
**Effects of the molecular weight of PEG and PCDTBT on the morphology of blend films.** AFM height images of the washed films correspond to blend films produced with different molecular weights of PEG: (**a**) 1, (**b**) 4.6, (**c**) 20, (**d**) 35, and (**e**) 100 kg/mol. (**f**) The plot of the height and density of PCDTBT aggregates against the *M*_n_ of PEG. AFM height images of the washed films correspond to blend films prepared with PCDTBT molecular weights of (**g**) 16.2 and (**h**) 30.0 kg/mol. (**i**) Cross-sectional profiles along the dash lines in (**g**,**h**). The size of each AFM image is 3 × 3 μm^2^.

**Figure 4 materials-14-05071-f004:**
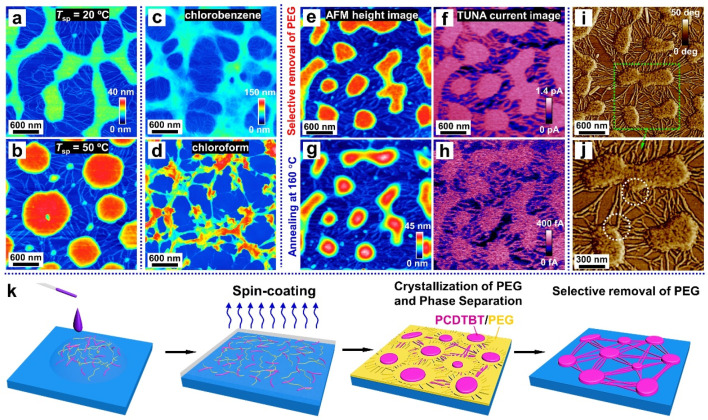
**Mechanism of the formation of the PCDTBT network.** AFM height images of the washed films correspond to blend films produced at different spin-coating temperatures (*T*_sp_): (**a**) 20 °C and (**b**) 50 °C. AFM height images of the washed films correspond to blends spin-casted from solutions with different solvents: (**c**) chlorobenzene and (**d**) chloroform. (**e**,**g**) AFM height images and (**f**,**h**) corresponding TUNA current maps of PCDTBT aggregates and nanofibers. (**i**,**j**) AFM phase images of PCDTBT aggregates and nanofibers. (**k**) Schematic representation of the formation of PCDTBT aggregates and nanofibers at various steps.

## Data Availability

The data presented in this study are available on request from the corresponding author.

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
