# Peer review of "Fabrication of PCDTBT Conductive Network via Phase Separation"

_materials, 2021, doi:10.3390/ma14175071_

Round 1
Reviewer 1 Report
Materials-1349030
Comments to the Editor
Manuscript number Materials-1349030
“Fabrication of PCDTBT Conductive Network via Phase Separation” presents the fabrication of a Poly[N-9''-hepta-decanyl-2,7-carbazole-alt-5-5-(4',7'-di-2-thienyl-2',1',3'-benzothiadiazole)] (PCDTBT) network consisting of aggregates and nanofibers in thin films, by phase separation of mixed PCDTBT and polyethylene glycol (PEG). The effect of blending conditions and processing conditions on the resulting phase separation morphologies of the blend films after a selectively washing procedure was studied.The phase separation structure transition from island to continuous structure occurred with the weight ratio of PCDTBT/PEG changed from 2:8 to 7:3. By increasing the solution concentration from 0.1 to 3.0 wt %, both PCDTBT height of aggregate and width of nanofiber increased. The film exhibited larger PCDTBT aggregate size, with the increase of the molecular weight of PEG. The electrical characteristic of PCDTBT network was measured by conductive AFM.
If the manuscript would have some minor revision before publishing (editing mistakes, the names-legend of the figures are a little too loaded, maybe it’s better to explain its in the text), it will be interesting for the readers of the Materials.
Reviewer 2 Report
Poly[N-9''-hepta-decanyl-2,7-carbazole-alt-5-5-(4',7'-di-2-thienyl-2',1',3'-benzothiadiazole)
CORRECT Poly[N-9′-heptadecanyl-2,7-carbazole-alt-5,5-(4′,7′-di-2-thienyl-2′,1′,3′-benzothiadiazole)]
Abstract: (PCDTBT) is a stable semiconducting polymer with high rigidity of molecular chains, which makes it difficult to organize into ordered structure, affecting the device performance.
Page 2 Abstract: the electrical characteristic of PCDTBT CORRECT: the electrical characteristics of PCDTBT
Page 3: the orientation of micro- and nano-scale ordered structures also play an important role CORRECT the orientation of micro- and nano-scale ordered structures also plays an important role
Page 3: to control the ordered structures of conjugated polymers as well as its properties CORRECT: o control the ordered structures of conjugated polymers as well as their properties
Page 3: reduced combinatorial entropy of mixed two types of polymer chains CORRECT reduced combinatorial entropy of mixing two types of polymer chains
Page 4: PCDTBT were blend with a well-studied model polymers PEG CORRECT: In this work PCDTBT was blended with a well-studied model polymer PEG
Page 4: The phase separation morphologies conductivity of PCDTBT will be investigated by using conductive atomic force microscopy (CAFM). CORRECT: Conductivity of the phase separated morphologies of PCDTBT was investigated by using conductive atomic force microscopy (CAFM).
Page 9: AFM height images the films CORRECT: AFM height images of the films
Page 9: a network consisting PCDTBT aggregates and nanofibers is revealed, while no nanofiber (especially nanofiber) is observed in Figure 2f1. CORRECT: a network consisting of PCDTBT aggregates and nanofibers is revealed, while no fibers (especially nanofibers) are observed in Figure 2f1.
Page 9: the similar morphology as compare with Figure CORRECT: the similar morphology as compared with Figure
Page 9: indicating no PCDTBT nanofiber exists CORRECT: indicating that no PCDTBT nanofibers exist
Page 9: crystalline polymer with relative high flexibility than PCDTBT – CORRECT: rystalline polymer with relative higher flexibility than PCDTBT
Page 9: Figure 2g-k perform the AFM height images CORRECT: Figure 2g-k shows the AFM height images
Page 10: It is imperative to point out that, the length of nanofiber decreased CORRECT: It is imperative to point out that the length of nanofiber decreased
Page 10: reducing the diffusion distance REMARK: How do you define here the diffusion distance?
Page 11: PEG with different molecular weights is blended PCDTBT (Mn = 16.2 kg/mol) due to mobility and compatibility is sensitive to the molecular weight of polymer34. REMARK: Logically and grammatically incorrect sentence, please rewrite.
Page 11: Figure 3a-e show CORRECT: Figure 3a-e shows
Page 11: are exhibit PCDTBT network CORRECT: exhibit PCDTBT networks
Page 12: With the increase of PEG molecular weight, the compatibility of two blend components decreases REMARK: is there any compatibility for low molecular weight PEG? The experimental work is based on phase separation driven by the incompatibility of PCDTBT and PEG.
Page 12: phase separation morphology CORRECT: phase separated morphology
Page 12: Figure 3g and 3h show CORRECT: Figures 3g and 3h show
Page 12: The molecular weight dependent of the density and width of PCDTBT nanofiber CORRECT: The molecular weight dependencies of the density and width of PCDTBT nanofiber
Page 13: thermodynamically state CORRECT: thermodynamic state
Page 13: the phase separation structure transition from continuous structure to island was observed CORRECT the transition from continuous to island structure was observed
Page 13: The evaporation rate of the solvent decides the time-span available COREECT: The evaporation rate of the solvent decides about the time-span available
Page 14: few nanofibers are observed in the chloroform sample CORRECT: few nanofibers are observed in the sample obtained from chloroform solution
Page 14: during the stage of spin-coating shorten time for phase separation, hindering PCDTBT molecules self-assembling into nanofibers. CORRECT during the stage of spin-coating with shorter time available for phase separation, thus hindering PCDTBT molecules self-assembling into nanofibers.
Line 14: After removal of PEG by washing the samples in acetone CORRECT After removal of PEG from the films by washing it out with acetone
Page 15: and subsequently formed nanofibers and aggregates CORRECT: and subsequently nanofibers and aggregates are formed.
Page 15: after selectively remove CORRECT: after selective removal
Page 15: sea-island phase separation structure CORRECT: sea-island phase separated structure
Page 16: no nanofiber was observed CORRECT no nanofibers were observed
Page 16: the conductivity of blend film was proved by conductive AFM- CORRECT: the conductivity of PCDTBT structures obtained from the blend films was proved by conductive
